# Ratoon Rice System of Production: A Rapid Growth Pattern of Multiple Cropping in China: A Review

**DOI:** 10.3390/plants12193446

**Published:** 2023-09-30

**Authors:** Wenge Wu, Zhong Li, Min Xi, Debao Tu, Youzun Xu, Yongjin Zhou, Zhixing Zhang

**Affiliations:** 1Rice Research Institute, Anhui Academy of Agricultural Sciences, Hefei 230031, China; lizhong021@126.com (Z.L.); ximin2015@126.com (M.X.); tudebao@aaas.org.cn (D.T.); xuyouzun@aaas.org.cn (Y.X.); zhouyongjin1111@163.com (Y.Z.); 2College of Agriculture, Anhui Science and Technology University, Chuzhou 239000, China; 3Key Laboratory of Crop Ecology and Molecular Physiology, College of Life Sciences, Fujian Agriculture and Forestry University, Fuzhou 350002, China; zhangzhixingfz@163.com

**Keywords:** ratoon rice, varietal characteristic, harvest, mechanization, eco-compensation

## Abstract

In this review, the significance of ratoon rice was introduced, and the research status and development trends of ratoon rice were also summarized. It is pointed out that mechanically harvested ratoon rice is the developing direction of future ratoon rice. On this basis, we analyzed the relationship between the yield of ratoon rice and many factors, such as variety characteristics, sowing date, water control, fertilizer, and many others. It is important to construct a comprehensive and practical evaluation system for rice regeneration that can provide a basis for high-yield cultivation of machine-harvested ratoon rice. At the same time, it is suggested that combining high-yield cultivation with the green ecological efficiency of rice can achieve better production and improve the quality of rice. Finally, some problems with ratoon rice development were put forward. An in-depth study on the rhizosphere biology and regulation techniques of ratoon rice and the effective ecological compensation mechanism increased the capacity and quality of ratoon rice. Further, the functioning of such research can enhance the planting area for ratoon rice and improve food security.

## 1. Introduction

Rice is an important food crop that provides for the food needs of nearly 60% of the world’s population. It has been estimated that global rice production will need to increase by 116 million tons (26%) by 2035 (from the 2010 production level) to meet the rising demand for rice [1]. Stabilizing and improving the total yield of rice has always been a hot issue. Previously, there were three main ways to increase the total crop yield: increasing the multiple cropping index, expanding arable land, and improving the yield per unit area [2]. However, climate change, water shortages, land shortages, arable land shortages, and labor shortages are also adverse factors that restrict the high yield of rice [3,4]. Therefore, increasing the multiple cropping index has become an important method to increase the total yield of rice [2]. Since the 21st century, rice production has shifted from simply pursuing yield to the multi-goal coordinated development of “high yield, high quality, high efficiency, ecology, and safety.” The change in production goals inevitably leads to a change in the crop production process. Ratoon rice is a resource-saving planting model that adopts particular cultivation and management measures to make the dormant buds on rice germinate into panicles after harvesting the main season of rice [5,6]. It has the characteristics of “seven decreases and two increases” (decrease of seed, labor, time, water, fertilizer, medicine, and rice field management, as well as an increase in rice quality and price) (Figure 1) [7,8]. Rice ratooning promotes agricultural supply-side structural reform, thereby implementing the national plan to reduce the area planted with double-cropping rice in the Yangtze River Basin, and helps to ensure China’s absolute grain security and optimize the production structure of grain [9]. Therefore, the research on the formation mechanism of yield and regulation technology of ratoon rice has crucial significance and reference value for promoting the sustainable development of ratoon rice, which is a significant demand to exert the rice cropping system to stabilize grain, increase income, and improve the contribution rate to regional food security. 

It is essential to enhance the theoretical basis of the ratoon rice production system to address the key challenges that are now limiting output and facilitate achieving the green and sustainable growth of the system. From the point of view of system theory, the ratoon rice production process is regarded as a system, namely the ratoon rice production system, and the synergistic relationship among various elements of the ratoon rice production system is studied deeply to tap the productive potential fully and effectively. This review compares the advantages and limitations of mechanical and manual harvesting of main-season rice and clarifies the importance of mechanical harvesting. On this basis, the factors of yield formation in mechanical harvest and three research cores of mechanical harvest ratoon rice were analyzed. The present review summarizes the latest research progress on integrated agronomic technologies to improve the productivity and sustainability of the ratoon rice system, and the problems and prospects in ratoon rice systems are also reviewed. 

## 2. Mechanical Harvesting of the Main Season Is the Key to the Development of Ratoon Rice

Countries such as Japan, Pakistan, the Philippines, Thailand, and the United States have all begun to develop ratoon rice cultivation techniques [10,11]. Ratoon rice has a history dating back 1800 years in China and was used as a disaster relief measure in the past. The development of ratoon rice has been repeated several times, and a new round of development began in 2011, stemming from the requirements of rice quality safety and farmers’ income increases. Low and unstable yields limit the development of ratoon rice technology [12]. However, with the continuous breakthrough of rice breeding and the continuous innovation of cultivation technology, the proportion of ratoon rice planting mode in the production system of traditional rice is increasing. Currently, there are two commonly used methods of ratoon rice harvesting: manual harvesting and mechanical harvesting [9]. In the early stage of ratoon rice cultivation, the harvesting method by artificial means with high segmental regeneration was mainly used to increase the adequate panicle number of ratoon rice by obtaining more axillary buds [13]. However, this planning mode of rice has been restricted due to its high production costs, such as labor intensity, and the fact that the area of artificial reaping has been sharply reduced. There is an urgent need to change and innovate the planting pattern of ratoon rice, one that is simple, high-yielding, and high-benefit. The rice planting model of machine-harvested ratoon rice has been increasingly welcomed by farmers and has made a contribution to the large-scale promotion of ratoon rice [9].

Compared with manual harvesting, mechanical harvesting in the main season of regenerated rice has some limitations and advantages (Table 1). The mechanized harvesting and production of ratoon rice significantly improve production efficiency, reduce labor costs, improve operation quality and production costs, and effectively compensate for its shortcomings. 

## 3. Factors Limiting the Yield of Machine-Harvested Ratoon Rice

The yield of mechanized ratoon rice is affected by many factors, such as variety characteristics [14], sowing date [15], pile height [16], water [17], and fertilizer control [18]. The shortage of heat and light resources, lack of strong regenerative varieties, considerable loss of the main-season rice harvester rolling, and high-yielding and high-quality technology resulted in low annual productivity and poor rice quality, which seriously restricted the development of ratoon rice industrialization (Figure 2). 

### 3.1. Strong Regenerative Varieties Are Lacking

The selection of excellent varieties is the basis of ratoon rice [19]. On the other hand, the viability of rice varieties is critical for judging the quality of rice varieties, and this is determined by the ability of buds to germinate into panicles. The fertility of rice is mainly affected by the genetic characteristics of rice varieties and the growing environment in the planting process, among which the genetic features of rice varieties play a dominant role [20]. There is still a lack of more ideal rice varieties with strong reproducibility that are suitable for mechanized harvesting. It is urgent to solve this problem by effectively screening superior rice varieties [9]. Nowadays, the standard method is to screen a large number of high-yielding varieties with strong fertility among the existing main-season rice varieties. According to statistics from 1979 to 2018, a total of 9563 rice varieties (6492 *indica* rice, 3024 *japonica* rice, and 47 *indica* rice) were approved at or above the provincial level in China, among which 4159 varieties were widely used in production [21]. It is too much work to screen out the adaptable regenerative varieties from such a large number of varieties, and it is difficult for the same varieties to achieve stable and high yields in many regions due to the different ecological environments in the different areas. It can be seen that this method lacks subjective initiative and foresight. In addition, axillary bud germination is the key to a high and stable rice yield in the ratooning season [22]. However, the mechanism of axillary bud germination is still unclear. It is therefore necessary to further study the cultivation technology and mechanism of tiller promotion in perennial root crops [9]. Given this, on the premise of ensuring the safety of ratooning season rice, it is necessary to adjust measures to local conditions by studying the head quarter season or regeneration of some agronomic traits such as chlorophyll content, photosynthetic capacity, and strength of the stem and root, as these are closely related to the generativity indicators [23]. To build a forecast for the appraisal standard of generativity to push the ratooning rice varieties screening work quickly and efficiently, it is beneficial to the popularization and application of ratoon rice varieties. 

### 3.2. Limited Temperature and Light Resources Makes It Difficult for Ratoon Rice to Mature Safely in the Ratooning Season

The sowing time of the main season of ratoon rice directly determines the harvesting time of the main season. It indirectly affects the germination of axillary buds into panicles in the ratoon season [24]. Previous studies found that the earlier the main season rice was sown, the higher the climatic productivity index was [24], and the longer the main season vegetative growth period was, the more effective the panicle number was and the higher the yield was. The later the sowing time of the main season of rice, the lower the yield, and even the yield of the ratoon season could not be obtained [25]. At the same time, if sowing too late, the safe full-heading stage of the ratooning season will be affected by low temperatures [23]. In actual production, producers have adopted various means to achieve seedling raising in advance, such as plastic film seedling raising [26], greenhouse seedling raising, and factory seedling raising [27]. In conclusion, according to the conditions of temperature and light resources in the planting area, the selection of ratoon rice varieties with a moderate growth period and sowing as early as possible is conducive to the growth of the reproducing season and increases yield (Figure 3). 

The first harvest time of ratoon rice affects the first harvest yield of ratoon rice and axillary bud germination, adequate panicle number, safe panicle, and yield [28]. Thus, the harvesting time of the main season determines the viability of rice varieties. Delaying harvesting shortens the ratooning season’s growth period, reducing rice yield in the ratooning season [29]. Furthermore, observation in the field has indicated that the best time to harvest main-season rice is when 80% of the ratoon buds protrude from the leaf sheath, resulting in the best rice yield in two seasons. Premature harvesting reduces output in the main season, while late harvesting leads to abnormal heading in the ratoon season. 

Under the existing planting patterns, high temperatures and heavy rainfall were the key meteorological factors affecting the high and stable yield in the main season. However, insufficient temperature and light resources in the growth period and low temperatures in the later period limited the yield potential in the ratooning season. Therefore, it is imperative to reasonably align the sowing date and harvesting date and use temperature and light resources to increase the yield of ratoon rice. 

### 3.3. The Main Season of Mechanical Rolling Caused a Serious Reduction in Production

There are still some problems with mechanical rolling weight and a significant loss rate in ratoon season, which seriously restrict the large-scale development of ratoon rice [23]. The effect of rolling in the main season can be effectively reduced by combining agricultural machinery and agronomic practices [30,31]. Lin et al. found that, compared with equal row spacing planting, wide and narrow row planting effectively increased the yield of main season rice, and this, combined with narrow-range crawler harvesting, could effectively reduce the rolling area of the crawler to the retained rice pile and reduce the influence of machine harvesting and rolling on the yield of the ratoon season [32]. The yield loss in the ratooning season caused by mechanical rolling during the first harvest can also be reduced through reasonable agronomic practices [33]. At present, the primary research focuses on the modes of “strong reproducible resistance to rolling cultivars” + “appropriate drought (soil moisture content 30%) harvesting” + “low node harvesting” + “mechanical travel path,” which effectively reduces the effect of mechanical rolling on the germination of ratoon buds and promotes the germination of ratoon buds into panicles. 

### 3.4. The Yield Was Unstable in the Main Season and Low in the Ratooning Season

As an essential organ for growth and development, the root system absorbs water and nutrients from the ground. It indirectly determines the performance of many agronomic traits of rice, such as yield, quality, stress resistance, and adaptability [26]. The root system of ratoon rice mainly consists of two parts: the old root surviving on the mother stem in the main season and the new root produced in the ratoon season, as well as a new root produced from a dormant bud root primordium on the mother stem of a main season rice post [34]. The premise of the high yield of ratoon rice is the effective combination of a strong main-season root system and a certain number of new roots. Moreover, rational regulation of water and fertilizer is an essential cultivation technique for ratoon rice production. Many studies have focused on the molecular responses of water and fertilizer regulation to the yield formation of ratoon rice and its physiological and ecological characteristics. Improper water and fertilizer management in main-season rice will lead to premature senescence of the plants or infection of rice stubs by diseases and insects, which will seriously affect the quality of rice stubs, thus affecting the survival and germination of axillary buds [23]. The researchers in southeast China promoted their low-pile ratooning rice cultivation technique. They emphasized nitrogen backwards and two times before baking in high-yield cultivation techniques: the purpose is through the control measures to make the water head of late-season rice roots remain of high vigor, leaves have no premature aging, and in harvest, rice piles of good quality and axillary buds have a high survival rate, a good regeneration rate, increase the effective panicles, and achieve a high yield and good quality [7]. Negalur et al. suggested that the rehydration time after the first harvest should be considered comprehensively with the height of the remaining pile to avoid the risk of regrowth bud necrosis caused by premature flooding and weed competition caused by late flooding in the ratooning season [35]. It is worth noting that the late harvest period of the first season of rice reduces the harvester’s mechanical crushing and does not cause the death of the ratoon buds [34]. Water management is vital to the cultivation of ratoon rice. After years of research, researchers in China came up with a formula for irrigating ratoon rice: (Main season) transplanting in shallow water, promoting tiller with little water, timely field drying, rehydration jointing, full water at booting, dry and wet grain, suitable for dry harvest. (Ratoon season): timely re-watering after cutting, shallow water regeneration, dry and wet grain, full dry harvest. 

There are two timings of N application for improving the grain yield of ratoon rice [34]. One is the N application for promoting bud development (N_bud_), which is applied during the period between the heading and harvest of the main crop. Another is the N application for promoting the development of ratoon tiller (N_tiller_), which is applied immediately after harvesting of the main crop [19,36]. The effects of these two fertilizers were, firstly, to increase the accumulation of dry matter in the early stages of the rice pile and promote its transport to ratoon tillers after harvest. Secondly, to expand the leaf area of ratoon tillers, increase the net photosynthetic capacity, and increase the dry matter accumulation of ratoon tillers In the process of rice grain filling, light contract compounds are basically supplied to the grain, and very few light contract compounds are allocated to the growth and development of ratoon buds, resulting in the loss of vigor of many ratoon buds due to nutrient deficiency. Therefore, the use of N_bud_ and N_tiller_ can effectively solve this problem. However, there are different opinions on the time and amount of application. Huang et al. applied N_bud_ and N_tiller_ in a 1:1 ratio (total N 180 kg·ha^−^^1^). The N_bud_ was applied 20 days after the early-season main rice heading, while the N_tiller_ was applied on the 3D after the main rice harvesting [7]. The reason may be related to climatic conditions and various characteristics in different regions. Therefore, applying bud-promoting fertilizer and seedling-raising fertilizer is essential to cultivating and managing ratoon rice. The application rate of fertilizer is related to the nitrogen application rate in the first season, hence the height of the remaining pile and the characteristics of rice varieties. Finally, once the local conditions are ideal, scientific fertilizer management has been observed to save money, improve efficiency, and reduce pollution [37].

## 4. Three Central Cores in the Research of Ratoon Rice

### 4.1. Compared with Main Season Rice and Late Heading Rice, Material Transport Was Higher in the Ratooning Season

The nutrients in the main-season rice plants came from photosynthesis in leaves, while the nutrients in the early growth stage of ratoon rice mainly came from the residual photosynthates in the mother stem [38]. The carbohydrate accumulation in the main-season rice post is the most important material guaranteed for the germination and early growth of ratoon seedlings [39]. Rice grain yield formation mainly depends on plant material production, accumulation, and grain distribution after heading [40]. However, in addition to the photosynthetic material produced in the ratoon season, the material for ratoon season rice is also obtained from the residue of the rice stump after the initial harvest. Xie’s researchers reported that head quarter in the new differentiation stage has the highest amount of dry matter accumulation, and the regeneration from a season to harvest season full panicle stage and ratooning season to mature accumulation were similar. The amount of dry matter accumulation and ratooning season were very significantly and positively correlated with yield, with the head of the high yield field season harvest index generally in 0.50–0.53. And the ratooning season can be as high as 0.67–0.70 [41]. Ratoon rice might have higher carbohydrate and nitrogen transportation efficiency than single rice crops. Ratoon rice is a typical flow-enhanced rice model. The NSC transport rate in the leaves and stem sheath of reproducing season rice is significantly higher than that of the main season late rice heading at the same time, and the NSC transport contributes as much as 10–18% to the yield. This characteristic of flow enhancement is the main physiological reason that the harvest index of ratoon season rice is higher than that of the same genotype late rice heading at the same time [7].

In conclusion, the activity of “source” (mainly rice stubs and old roots) and “sink” (mainly axillary buds and new roots) after harvesting of main-season rice and their substance transport determined the germination and growth of ratoon axillary buds [42]. However, the second material transport of the new “source” organs (ratoon roots, ratoon stems, and ratoon leaves) to the grain (“sink”) after the development of ratoon axillary buds is the fundamental factor that determines the yield formation of ratoon rice. 

### 4.2. Compared with Main Season Rice and Late Heading Rice, the Rice Quality in the Ratoon Season Was Better

In recent years, with the rapid development of the social economy, people’s living standards have greatly improved, and the requirements for good-quality rice are naturally becoming higher and higher [7]. The expensive rice on the market requires good appearance, taste, and aroma [43]. Improving rice quality has become a key problem limiting the development of the rice industry [9,23]. Compared with the traditional planting methods of single and double cropping rice, ratoon rice often has the advantages of good taste and quality [43], which could be a determinant factor in economic returns for farmers [44]. A recent study demonstrated that ratoon rice significantly improved the physicochemical properties and textural characteristics of cooked rice as compared to the single-cropping main crop, showing an increase in mean amylose content from 14.85% to 19.92% and a decrease in mean protein content from 8.11% to 5.6% [45]. The protein content of more than 8% in rice significantly reduced the taste of rice [46,47]. In addition, the appearance quality of ratoon rice was also significantly better than that of main-season rice, especially since the chalkiness rate was significantly reduced [48]. Many studies have demonstrated critical roles for amylose and protein in improving the grain quality of ratoon rice, including cooking, eating, nutritional, and textural properties. The quality of rice depends largely on the influence of environmental temperature. The ambient temperature change directly affects rice quality formation by affecting the accumulation of amylose and protein [49]. Interestingly, the average daily temperature during grain filling differed between main-season rice and ratooning rice. The better grain quality of ratoon rice may be due to the lower accumulated temperature during the growth period than that of main crop or double crop rice [50]. The low temperature in the ratooning season prolonged rice’s ripening period, promoting rice’s grain-filling process and quality [48]. In addition, the harvester’s rolling damage and the main season’s rice disease may lead to poor grain quality in the regenerating season. The ratoon rice was planted on the basis of the main-season rice, and the environmental factors are relatively stable. In contrast, main-season rice planting was often affected by climate change (drought and heavy rainfall), and these environmental factors might have a certain impact on the quality of rice. 

In conclusion, the investigation into the quality of ratoon rice holds profound significance in enhancing its palatability, nutritional efficacy, and market viability. Furthermore, it serves as an instrumental measure in securing our food supply and fostering the sustainable progression of agricultural practices.

### 4.3. Compared with Main Season Rice and Late Heading Rice, Ratoon Rice Had Lower Greenhouse Gas Emission

It is reported that rice fields are important sources of greenhouse gas emissions [51], and different rice varieties, water and fertilizer regulations, and rice cropping patterns all have impacts on greenhouse gas (GHG) emissions [52]. The N_2_O emission per unit yield of ratoon rice (2.06 kg·t^−1^) was also significantly lower than that of single-cropping rice (3.02 kg·t^−1^). The global warming potential (GWP), a comprehensive CH_4_ and N_2_O emissions index, was calculated, and it was found that the N_2_O emission of ratoon rice was 43.0% lower than that of single-cropping rice [53]. Song’s research found that the CH_4_ and N_2_O emissions from the ratoon rice seasons accounted for 23–24% and 10–27% of total emissions from the main rice crop and ratoon rice seasons, respectively, due to the low temperature, short growth period, and low aboveground biomass during the ratoon rice season [54]. Previous studies found that ratoon rice under plastic film can improve the net ecosystem economic budget and reduce greenhouse gas intensity, potentially leading to a win-win situation for the environment and growers in water-stressed regions [53,55].

Therefore, the development of ratoon rice cultivation has great potential to reduce greenhouse gas emissions in the process of food production [56]. However, there are few reports on the total direct and indirect greenhouse gas emissions (i.e., carbon footprint) during the production of ratoon rice and double-rice or the ratooning season of ratoon rice and the same heading, late rice [55], and the influencing factors for the difference in emissions are also worth further investigation [54]. The importance of research on reducing greenhouse emissions from ratoon rice is reflected in reducing greenhouse gas emissions, protecting the environment, and improving agricultural sustainability. Further study and promotion of ratoon rice cultivation methods can contribute to climate change mitigation and sustainable agricultural development. 

## 5. Conclusions and Prospects

Rice researchers in different regions of China have accumulated rich experience and formed many regulations and techniques suitable for local cultivation after long-term research and exploration (Table 2). Machine-harvested ratoon rice has become the latest development of future ratoon rice. It is necessary to build a comprehensive and practical evaluation system for rice regenerative capacity to provide a basis for high-yield cultivation of machine-harvested ratoon rice. On this basis, it was suggested that the lack of strong regenerative varieties, shortage of temperature and light resources, mechanical rolling, and unstable yield were the important factors limiting the yield formation of the mechanical harvesting of ratoon rice. At the same time, ratoon rice’s three core points (material transport, rice quality, and greenhouse gas emissions) were introduced. 

The state puts food security first, while farmers pay attention to the improvement of economic benefits. Thus, coordinated efforts are required in order to strike a balance between increasing grain production and increasing farmers’ income. Therefore, policy support is needed to explore the ratoon rice industry’s cultural connotation and quality value and improve industrial benefits through brand building. In view of this, it is necessary to study the rhizosphere biology and regulation technology of regrowing rice, promote the scientific and standard planting of regrowing rice, and effectively carry out this study of the ecological compensation mechanism and function of regrowing rice. All these efforts can stabilize and develop the planting area for regrowing rice and ultimately attain food security not only in China but across the globe. 

## Figures and Tables

**Figure 1 plants-12-03446-f001:**
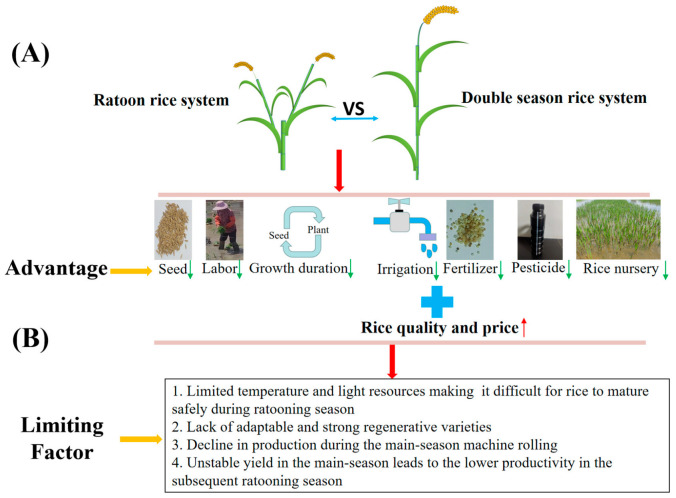
The comparison between the ratoon rice system and the double-season rice system. (**A**) The advantage of the ratoon rice system compared with the double-season rice system; (**B**) The limiting factor of the ratoon rice system.

**Figure 2 plants-12-03446-f002:**
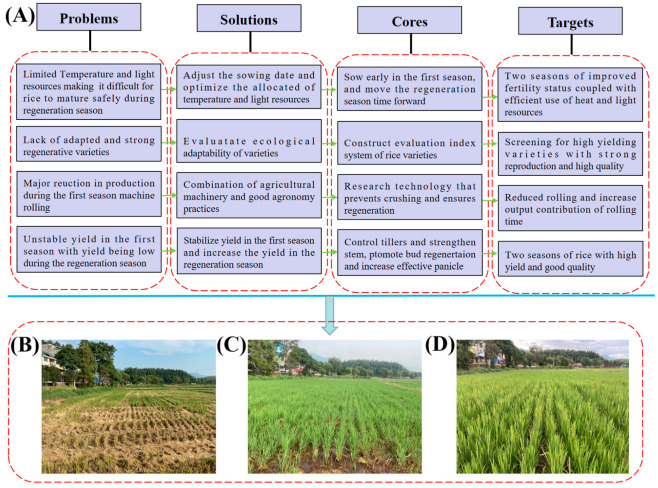
Research status and development goal of ratoon rice. (**A**) Technology roadmap; (**B**–**D**) A pictorial illustration of ratoon rice; (**B**) The ratoon crop at 5 days after the harvesting of the main crop; (**C**) The ratoon crop at 15 days after harvesting of the main crop; (**D**) The ratoon crop at the heading stage.

**Figure 3 plants-12-03446-f003:**
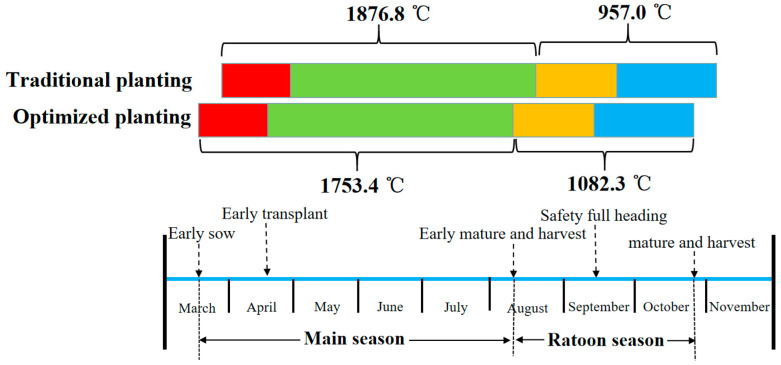
Schematic diagram of the ratoon rice growth process (taking the area along the Yangtze River in Anhui Province (Hefei) as an example).

**Table 1 plants-12-03446-t001:** The advantages and limitations in mechanical harvesting and manual harvesting.

	Mechanical Harvesting	Manual Harvesting
Advantages	Mechanical harvesting efficiency is high and can quickly harvest a large area of harvesting work, saving human resources. Mechanical harvesting can improve the quality of crops and reduce loss and waste. Mechanical harvesting can complete the harvesting work quickly, which is conducive to timely crop treatment and reduces the impact of weather and other factors on crops. Mechanical harvesting can realize the modernization and automation of agriculture and improve the efficiency and competitiveness of agricultural production.	Manual harvesting has high flexibility, can operate in complex farmland environments, and has strong adaptability. Manual harvesting is less damaging to the field and protects the integrity of the soil and crops. Artificial harvesting can provide employment opportunities, promote the rural economy, and increase farmers’ incomes. Manual harvesting can maintain the farmland’s landscape and ecological environment, which is conducive to the sustainable development of farmland.
Limitations	Investing higher funds to purchase and maintain agricultural machinery and equipment is necessary, which puts greater economic pressure on farmers. Machinery may encounter various problems in different farmland environments, such as mudponds, paddy fields, mountains, and other terrain constraints. Mechanical harvesting can cause damage to fields, especially in wetland conditions, which tend to leave deep ruts. Skilled operation techniques and maintenance knowledge are required to ensure the normal operation of machinery.	Manual harvesting needs a lot of manpower input, labor intensity, and high demand for labor resources. The speed of artificial harvesting is relatively slow and cannot adapt to the rapid harvesting needs of large farmland areas. Manual harvesting is prone to accidental crop injury and loss, making it challenging to control crop quality. Manual harvesting will be limited by the personnel’s technical level and physical condition and is unsuitable for long-term continuous work.

**Table 2 plants-12-03446-t002:** General principles of ratoon rice development.

General Principles	Describe
Stabilizing food is a top priority	On the premise of ensuring the safety of reproduction rice in two seasons, according to the variation characteristics of regional temperature and light resources, the optimization model of the growth process of machine-harvested reproduction rice was proposed, which was premature beating, early planting, and early harvest in the main season.
Choose good varieties	Scientific selection of high-quality varieties with strong regenerative power and high yield is suitable for mechanization promotion and optimization of nitrogen fertilizer management, coupled with wet irrigation, to enhance the material accumulation in the stem sheath in the middle and late growth periods, strengthen stem aging, promote grain filling, improve the activity of axillary buds at the base, and promote the germination of regenerative buds after the first harvest.
Reduce rolling	The main season of rice harvesting significantly reduced the budding rate during the ratooning season. Reducing the effect of mechanical rolling on the growth of regenerated buds is the key to increasing rice yield in the regenerated season.
Green security	To develop green, circular, and low carbon clean and green ecological cultivation modes of machine-harvested reproducing rice, solve the problem of increasing yield without increasing income, and realize sustainable development.

## Data Availability

All data generated during this study is included in this published article, and the raw data used or analyzed during the current study is available from the corresponding author on reasonable request.

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
