# Peer review of "Ratoon Rice System of Production: A Rapid Growth Pattern of Multiple Cropping in China: A Review"

_plants, 2023, doi:10.3390/plants12193446_

Round 1
Reviewer 1 Report
The paper covers a relevant and scientifically interesting topic. But its writing and structure are unorganized, it is a hard read. I addition the writing is sloppy, which can be seen by using the same heading again and again without any connection to the presented content. Somehow the impression is created, that the authors report the minutes of a meeting of Chinese scientists debating ratoon rice. It is not the purpose of an international scientific journal to solve Chinese rice productivity prolems, but to present the science that can be connected to it.
For a renewed submission, the paper should give a proper explanation of ratoon rice production system compared to double cropped paddy rive, including all management activities.
The problems (yield, management?) and advantages (save water, costs?) should be presented in enough details and not in form slogans of "5 y smaller, 2 x larger". In the following, there should be a clear structure on how the open scientific questions of ratoon production can be tackled and what has been achieved so far, with a clear structure like: harvest, regrowth, canopy management... etc, etc.
The authors should eventually contact the editorial office before a resubmission.
the manuscript has many English language issues with respect to grammar (singular/plural, tense) but also choice of technical terms.
The paper need a thorough revisison with respect to this issue.
Author Response
1.The paper covers a relevant and scientifically interesting topic. But its writing and structure are unorganized, it is a hard read. I addition the writing is sloppy, which can be seen by using the same heading again and again without any connection to the presented content. Somehow the impression is created, that the authors report the minutes of a meeting of Chinese scientists debating ratoon rice. It is not the purpose of an international scientific journal to solve Chinese rice productivity problems, but to present the science that can be connected to it.
Reply: Thanks for constructive comments. We’ve addressed all the above-mentioned issues in the revised manuscript. Revised manuscript is now grammatically, technically and logically well organized and quality of the manuscript is enhanced.
2. For a renewed submission, the paper should give a proper explanation of ratoon rice production system compared to double cropped paddy rive, including all management activities.
Reply:Thanks for constructive comments. The revised manuscript has been modified accordingly.
3.The problems (yield, management?) and advantages (save water, costs?) should be presented in enough details and not in form slogans of "5 y smaller, 2 x larger". In the following, there should be a clear structure on how the open scientific questions of ratoon production can be tackled and what has been achieved so far, with a clear structure like: harvest, regrowth, canopy management... etc, etc.
The authors should eventually contact the editorial office before a resubmission.
Reply:Thanks for critical review. We’ve substantially modified the current manuscript. Revised manuscript has been improved through more logical and constructive approach.
4.the manuscript has many English language issues with respect to grammar (singular/plural, tense) but also choice of technical terms.The paper need a thorough revisison with respect to this issue.
Reply: Thanks for critical review. The manuscript’s grammatical and English language errors are carefully revised by a native English language speaker. The quality of revised manuscript is improved now.
Reviewer 2 Report
The review provides valuable insights into the significance of ratoon rice and presents a comprehensive summary of the current research status and development trends in this field. The emphasis on mechanically harvested ratoon rice as the future direction is well-founded. The analysis of factors affecting ratoon rice yield, including variety characteristics, sowing date, water control, and fertilization, is commendable. The suggestion to establish a practical evaluation system for rice regeneration is crucial for achieving high-yield cultivation of machine-harvested ratoon rice. Furthermore, the recommendation to integrate high-yield cultivation with green ecological efficiency is a promising approach for enhancing production and rice quality. The review also highlights important areas for further investigation, such as rhizosphere biology, regulation techniques, and ecological compensation mechanisms, which are essential for stabilizing and expanding the planting area of ratoon rice and ensuring food security. Overall, this review contributes significantly to the advancement of ratoon rice research and development. I recommend this review article to be accepted.
I recommend this review article to be accepted.
Author Response
Reply: Thank you for your kind views for our manuscript.
Reviewer 3 Report
Dear authors and editor, I was pleased to know about the Ratoon Rice system of production: a rapid growth pattern of multiple cropping in China. A review.
The authors have done a job that needs to be improved
Introduction
It is well done, but not describes the current state of the research. 1.Recommendations are about improving the definition of the research scope.
2. Also, conclude the paragraph of the introduction with the aim of the work. What are the contributions of the study? It seems not clearly defined in the introduction.
3. The basic objectives and the research methodology adopted to fulfill the objectives should be included, preferably under a separate heading.
4. The importance and practical application of the work needs to be highlighted, preferably under a separate heading.
5.The conclusion should be focused on the primary research findings
Best regards
Author Response
1.Recommendations are about improving the definition of the research scope.
Reply:Thank you for kind suggestion. We've revised the scope of the study. Firstly, explained the significance of ratooning rice development. Based on this, mechanical harvesting recommended as the key of ratooning rice development. At the same time, a limiting factor in the development of ratooning rice was pointed out. Finally, the core research point of mechanized production of ratooning rice was put forwarded.
2.Also, conclude the paragraph of the introduction with the aim of the work. What are the contributions of the study? It seems not clearly defined in the introduction.
Reply: Thanks for your comments. In the revised manuscript we’ve added a conclusive paragraph in the “Introduction” section. (New Line 67-74).
3.The basic objectives and the research methodology adopted to fulfill the objectives should be included, preferably under a separate heading.
Reply: Thanks for pointing. We’ve detailed explained it in the revised manuscript under the section “Conclusion and Prospects”.
4.The importance and practical application of the work needs to be highlighted, preferably under a separate heading.
Reply: Thank you for critical review. We’ve explained it in the revised manuscript. (New Line 303-312 and line 334-338).
5.The conclusion should be focused on the primary research findings
Reply: Thanks for critical review. The conclusion section has been rectified accordingly.
Reviewer 4 Report
The manuscript entitles “Ratoon rice system of production: a rapid growth pattern of multiple cropping in China. A review” is a concise and interesting contemporary topic of study on food security. In this article, the significance of ratoon rice is discussed along with its research status, development trends, and potential future growth directions. At the same time, certain recommendations for combining rice production with high-yield and ecologically conscious practices were made. However, some details must be clarified and some things must be discussed in greater detail for the benefit of the readers.
1. Line 23-24, where is the rhizosphere biology and regulation technology of ratoon rice mentioned in the article?
2. Line 33-34, please provide specific references for the statement.
3. What are the limitations of the mechanical harvesting of ratoon rice? Please compare and explain in detail the limitations and advantages of mechanical harvesting of ratoon rice.
4. As a review article, it may be more references to summarize your conclusions.
There exist some mistakes in expressions and words. For example, in Figure 1, there are spelling mistakes. Thorough checks are necessary to improve this manuscript.
Author Response
1.Line 23-24, where is the rhizosphere biology and regulation technology of ratoon rice mentioned in the article?
Reply: Thanks for pointing. We’ve mentioned it in the heading “Conclusion and Prospects”, It is need of the hour to develop such regulation technology for enhancing the efficiency of the ratoon-rice cropping system.
2.Line 33-34, please provide specific references for the statement.
Reply: Thank you for pointing. The reference has been added to the statement. (New Line 33-34).
3.What are the limitations of the mechanical harvesting of ratoon rice? Please compare and explain in detail the limitations and advantages of mechanical harvesting of ratoon rice.
Reply: Thanks for critical review. We’ve addressed the above mentioned question in the revised manuscript and explained in detail in modified “Table 1”. (New Line 75-102).
4.As a review article, it may be more references to summarize your conclusions.
Reply: Thanks for your kind suggestion. We added new and recent references in the revised manuscript.
5.There exist some mistakes in expressions and words. For example, in Figure 1, there are spelling mistakes. Thorough checks are necessary to improve this manuscript.
Reply: Thanks for critical review. The manuscript’s grammatical and English language errors are carefully revised by a native English language speaker. The quality of revised manuscript is improved now.
Reviewer 5 Report
This article is meaningful and provides a corresponding overview about the importance, current advances, and future trends on the subject of ratoon rice. However, several significant problems fellowing listed should be corrected.
1. No legend in the figure 1
2. The title 2.3 (Line 162)is identical with title 2.4(Line 177)
3. Same title between table 1 and 2
4.The title 3.1(Line 235),3.2(Line 264)and 3.3(Line 294)are exactly the same
5.Part 3.2 and 3.3 does not implicit the importance why extra research is necessary
6.The terminology, such as reproduce rice recycle rice, regrowing rice ratoon rice,should be reconsidered
7.On account of frequent grammar error, this manuscript needs to be carefully revised, and read through by a native speaker.
minor issues:
1.(Line 77)typo control
2.The title of Figure 3 should not be all capitalized.(Line144-145)
3.What does ring-rice mean?(Line327)
On account of frequent grammar error, this manuscript needs to be carefully revised, and read through by a native speaker.
Author Response
1.No legend in the figure 1
Reply: Thanks for pointing. The figure legends have been added in the revised manuscript and explained in the description.
2.The title 2.3 (Line 162)is identical with title 2.4(Line 177)
Reply: Thanks for pointing. We’ve modified and rephrased it.
3.Same title between table 1 and 2
Reply: Thanks for pointing out the mistake. We have modified in the revised manuscript. Now both tables have different titles and also improved content of tables.
4.The title 3.1(Line 235),3.2(Line 264)and 3.3(Line 294)are exactly the same
Reply: Thank you for pointing out. We’ve modified it in the revised manuscript.
5.Part 3.2 and 3.3 does not implicit the importance why extra research is necessary
Reply: Thank you for critical review. We’ve explained it in the revised manuscript. (New line 303-312 and line 334-338)
6.The terminology, such as reproduce rice recycle rice, regrowing rice ratoon rice,should be reconsidered
Reply: Thank you for pointing out. This terminology mistake has been modified in revised manuscript. It has been modified to “ratoon rice”.
7.On account of frequent grammar error, this manuscript needs to be carefully revised, and read through by a native speaker.
Reply: The manuscript’s grammatical and English language errors are carefully revised by a native English language speaker. The revised manuscript is of good quality and well organized now.
minor issues:
1.(Line 77)typo control
Reply: Thank you for pointing out. It has been modified.
2.The title of Figure 3 should not be all capitalized.(Line144-145)
Reply: Thanks for kind suggestion, I’ve modified it accordingly. ( New line 159-160)
3.What does ring-rice mean?(Line327)
Reply:It was typing mistake. It is ratoon-rice. It is modified now.
4.On account of frequent grammar error, this manuscript needs to be carefully revised, and read through by a native speaker.
Reply: The manuscript’s grammatical and English language errors are carefully revised by a native English language speaker. The revised manuscript is of good quality and well organized now.